

# HEMF: an adaptive hierarchical enhanced multi-attention feature fusion framework for cross-scale medical image classification

Jingdong He, Qiang Shi, Jun Ma, Dacheng Shi and Tie Min

School of Cyber Science and Engineering, Southeast University, Nanjing, Jiangsu, China

## ABSTRACT

Medical image classification is essential for contemporary clinical diagnosis and decision support systems. However, medical images generally have similar inter-class features and complex structure patterns, making it a challenging task. While both local and global features are critical for noise reduction and discriminative pattern extraction in medical images, conventional approaches exhibit limitations. Specifically, convolutional neural networks (CNNs) focus on local features extraction but lack a comprehensive understanding of global semantic. Conversely, vision transformers (ViTs) can model long-range feature dependencies but may cause disruption to local features. To address these limitations, we propose Hierarchical Enhanced Multi-attention Feature (HEMF), an adaptive hierarchical enhanced multi-attention feature fusion framework to synergistically extract and fuse multi-scale local and global features. It comprises two core components: (1) the enhanced local and global feature extraction modules to extract multi-scale local and global features in parallel; (2) the hierarchical enhanced feature fusion module integrating a novel attention mechanism named Mixed Attention (MA) and a novel inverted residual block named Squeezed Inverted Residual Multi-Layer Perceptron (SIRMLP) to effectively fuse multi-scale features. Experimental results demonstrate that with nearly minimal model parameters compared to other advanced models, HEMF achieves the accuracy and F1-score of 87.34% and 78.89% on the ISIC2018 dataset, 87.03% and 87.02% on the Kvasir dataset, and 82.26% and 82.20% on the COVID-19 CT dataset, which are the state-of-the-art performance. Our code is open source and available from https://github.com/Esgjgd/HEMF.

## INTRODUCTION

Medical image classification research focuses on categorizing medical images into disease-specific patterns to identify specific diseases. As a core technology for clinical diagnosis and decision support systems, medical image classification is essential to computer-aided diagnosis, image mining, and retrieval in healthcare.

Recently, convolutional neural networks (CNNs) have exhibited remarkable efficacy in numerous medical image classification tasks (*Shen et al., 2017*; *Koitka & Friedrich, 2016*; *Tajbakhsh et al., 2016*; *Hassan et al., 2020*). Although CNNs focus on local features, they

Corresponding author
Jun Ma, junma@seu.edu.cn

often lack a comprehensive understanding of global semantic information, which hinders their performance. Similar to transformer, vision transformer (ViT) (*Dosovitskiy et al., 2020*) splits the image into patches with position embedding to capture long-range feature dependencies and global semantic information. Unlike CNNs, pure ViT-based methods usually pay insufficient attention to local features and even disrupt low-level details (*Azad et al., 2024*). This limitation highlights the necessity of integrating CNN and ViT paradigms.

CNN-transformer based methods have emerged to synergize local feature extraction with global context modeling. Recent studies demonstrate that such integration contributes to reducing noise in medical images (*Dai, Gao & Liu, 2021*; *Zhang, Liu & Hu, 2021*; *Chen et al., 2022*). Furthermore, hierarchical fusion of multi-scale features has also proved to be essential for medical image analysis, (*Wang et al., 2023*; *Huo et al., 2024*; *Li, 2023*). These works indicate the promising direction of medical image classification research.

Motivated by the success of previous studies (*He et al., 2016*; *Liu et al., 2022*; *Sandler et al., 2018*; *Fan et al., 2024*) and to mitigate the limitations of model complexity and insufficient feature characterization capabilities, we propose an adaptive Hierarchical Enhanced Multi-attention Feature fusion framework named HEMF. Specifically, we design an enhanced local feature (ELF) extraction and enhanced global feature (EGF) extraction module, which allow the parallel extraction of multi-scale enhanced features. To fuse the multi-scale local spatial features and global semantic information, a hierarchical enhanced feature (HEF) fusion module is proposed. Within this module, we design a novel attention mechanism named Mixed Attention (MA). It includes a multi-head external attention, a spatial attention as well as a channel attention to learn multiple salient feature relationships among different samples and further focus on significant features and salient spatial regions. Moreover, we also introduce a novel inverted residual block named Squeezed Inverted Residual Multi-Layer Perceptron (SIRMLP) in this module to learn the high-dimensional representations of the fused features with a relatively low cost. Overall, unlike the previous works, our approach enables more effective extraction and fusion of both global and local features, resulting in state-of-the-art performance on three real-world medical image datasets. Remarkably, our approach achieves this performance with fewer model parameters and lower floating point operations (FLOPs). As a general module, the proposed HEF fusion module can be further adopted in other downstream tasks for efficient feature fusion. To improve readability, Table 1 provides a list of all acronyms used in this article.

In conclusion, the primary contributions of our article are summarized as follows:

- A novel adaptive hierarchical enhanced multi-attention feature fusion framework for cross-scale medical image classification named HEMF is proposed. Enhanced local feature (ELF) and enhanced global feature (EGF) extraction modules are designed to capture multi-scale local spatial features and global semantic information, providing a robust semantic representation for classification.

**Table 1 List of acronyms used in this article.**

| List of acronyms | |
|---|---|
| CNN | Convolutional neural network |
| ViT | Vision transformer |
| ELF | Enhanced local feature |
| EGF | Enhanced global feature |
| HEF | Hierarchical enhanced feature |
| MA | Mixed Attention mechanism |
| MHEA | Multi-Head External Attention mechanism |
| SA | Spatial Attention mechanism |
| CA | Channel Attention mechanism |
| SIRMLP | Squeezed Inverted Residual Multi-Layer Perceptron |
| FLOPs | Floating point operations |
| ML | Machine learning |
| SVM | Support vector machine |
| KNN | K-nearest neighbor |
| LN | Layer normalization |
| BN | Batch normalization |

- A hierarchical enhanced feature (HEF) fusion module is proposed to effectively fuse multi-scale enhanced local and global features. Within the module, a Mixed Attention mechanism and a novel inverted residual block SIRMLP are designed to learn multiple salient feature relation of different samples. It can further focus on key features and salient spatial regions, as well as learn high-dimensional representations of the fused features at low computational cost.
- The proposed HEMF model achieves state-of-the-art performance on three real-world medical datasets with nearly minimal model parameters compared to other advanced models. Specifically, HEMF reduces about 47.86% model parameters and 8.75% FLOPs compared to the advanced model HiFuse_Base.

## RELATED WORK

This section presents a comprehensive survey of medical image classification and related methodologies, organized into three paradigms: methods based on traditional machine learning (ML-based), CNN-based approaches, and transformer-based approaches. A comparative analysis of their strengths and weaknesses is provided in Table 2, along with related works in each category.

### ML-based methods

To classify medical images, traditional machine learning approaches such as support vector machine (SVM), K-means and K-nearest neighbor (KNN) rely on the extraction of relevant features such as color, texture, and shape, or feature combinations derived therefrom. *Rani, Mittal & Ritambhara (2016)* use SVM classifiers to detect and classify

**Table 2 Comprehensive literature review of three methodologies.**

| Methodology | Strength | Weakness | Related works |
|---|---|---|---|
| ML-based | Fewer computational resources, high-throughput processing, robust performance on small datasets, strong interpretability | Rely on manual feature engineering, limited nonlinear modeling capacity, incomplete high-dimensional features, the curse of dimensionality | *Rani, Mittal & Ritambhara (2016), Iwahori et al. (2015), Manju, Meenakshy & Gopikakumari (2015)* |
| CNN-based | Effective image processing, local feature modeling, robustness to spatial shift | Lack of global feature modeling, time-consuming in transfer learning, overfitting on small dataset, weak interpretability | *Shen et al. (2017), Koitka & Friedrich (2016), Flayeh & Douik (2024), Tajbakhsh et al. (2016), Hassan et al. (2020)* |
| Transformer-based (integrated with other architecture) | Both global and local feature modeling, high performance | Require more training data, complex model, time-consuming, overfitting on small datasets, weak interpretability | *Dai, Gao & Liu (2021), Zhang, Liu & Hu (2021), Dai et al. (2021), Chen et al. (2021), Wang et al. (2024), Huo et al. (2024), Li (2023)* |

focal liver lesions. *Iwahori et al. (2015)* use Hessian filter and K-means++ to detect the polyp region. *Manju, Meenakshy & Gopikakumari (2015)* extract texture features and use KNN classifier to diagnose prostate disease. These shallow methods typically require fewer computational resources compared to deep learning methods, making them feasible for deployment on low-power clinical hardware. Training converges faster due to simpler architectures, suitable for scenarios with limited time or energy budgets. They often outperform deep learning methods when labeled training data is scarce, as they rely on handcrafted features rather than data-hungry feature learning. Besides, they often provide strong interpretability, aligning with clinical demand for transparency in diagnosis. However, such approaches also possess inherent limitations. Performance heavily depends on manual feature engineering, requiring domain expertise to design optimal descriptors such as texture and shape. Handcrafted features may fail to capture complex spatial hierarchies in high-dimensional images. Linear models struggle with nonlinear feature interactions which are common in medical images. Furthermore, they are susceptible to the curse of dimensionality when dealing with high-dimensional image data.

## CNN-based methods

Convolutional neural networks (CNNs) are designed to extract salient features from structured data as multi-dimensional arrays, enabling a highly effective application on image data processing. Convolutional operations inherently capture local receptive fields, ideal for detecting fine-grained pathological features. Weight sharing and pooling operations provide robustness to small spatial shifts, which is critical for anatomical structure alignment. Their inherent ability to automatically learn relevant features has contributed to their effectiveness and widespread use in medical image classification. *Shen et al. (2017)* propose a multi-crop convolutional neural network, employing a novel multi-crop pooling strategy to automatically extract salient lesion information. *Koitka & Friedrich (2016)* utilize a single CNN for skin lesion classification, achieving end-to-end training with only images and labels as input. *Flayeh & Douik (2024)* provide a lightweight model based on autoencoder and CNNs to detect breast cancer. Besides, several studies

conduct transfer learning on CNNs, pre-training on a substantial *corpus* of medical images and then fine-tuning for the downstream classification task. *Tajbakhsh et al. (2016)* investigate the performance of deep CNNs trained from scratch compared to pre-trained CNNs with layer-wise fine-tuning. *Hassan et al. (2020)* utilize a pre-trained ResNet50 model for feature extraction and optimization, and then classify with linear discriminator. However, CNNs primarily focus on local spatial feature details, often lack a comprehensive understanding of global semantic information, which is crucial for performance enhancement. Furthermore, transfer learning methods can be time-consuming and necessitate substantial datasets for pre-training. Fine-tuning pre-trained models on small datasets risks overfitting, especially for rare diseases. Besides, Black-box decision-making may undermine clinical trust; while saliency maps such as Gradient-weighted Class Activation Mapping (Grad-CAM) (*Selvaraju et al., 2017*) and Grad-CAM++ (*Chattopadhay et al., 2018*) provide *post-hoc* explanations, they lack causal reasoning aligned with medical knowledge. End-to-end training may ignore established some features, limiting physician acceptance.

## Transformer-based methods

In recent years, methods based on self-attention mechanisms, such as vision transformer (ViT) (*Dosovitskiy et al., 2020*), have demonstrated remarkable performance in medical image classification tasks. ViT splits the input image into patches with position embedding to capture long-range feature dependencies and global semantic information. However, it usually does not pay enough attention to local features and can even disrupt low-level details (*Azad et al., 2024*). Moreover, it typically requires more training data than CNNs to converge, exacerbating challenges in rare disease classification.

CNN-transformer methods integrate the advantages of CNNs and ViTs, enabling the extraction and fusion of both local and global features. *Dai, Gao & Liu (2021)* use CNNs and ViTs to classify parotid gland tumors and knee injuries. *Zhang, Liu & Hu (2021)* fuse transformers and CNNs for medical image segmentation. *Dai et al. (2021)* combine the strengths from transformers and CNNs for image classification. Several studies investigate integrating transformers with other architectures or mechanisms for specific tasks. *Chen et al. (2021)* fuse transformers and U-net architecture for medical image segmentation. *Wang et al. (2024)* combine transformer with multi-granularity patch embedding and self attention for medical time-series classification. However, the aforementioned works perform feature fusion at a single scale of the samples, failing to integrate multi-scale representations, which hinders their ability to effectively capture the hierarchical and complex structures present in medical images. Recently, several studies investigate hierarchically extracting and fusing local and global features to suppress the incorporation of noise. HiFuse (*Huo et al., 2024*) designs a hierarchical feature fusion module named Hierarchical Feature Fusion (HFF) to fuse local and global features from the multi-layer encoders. However, its feature characterization capabilities can be further enhanced, and the model also exhibits a relatively large number of parameters and FLOPs. Based on HiFuse, CAME (*Li, 2023*) introduces an external attention mechanism to learn the relation of different samples. However, its attention mechanism is designed as the fourth branch

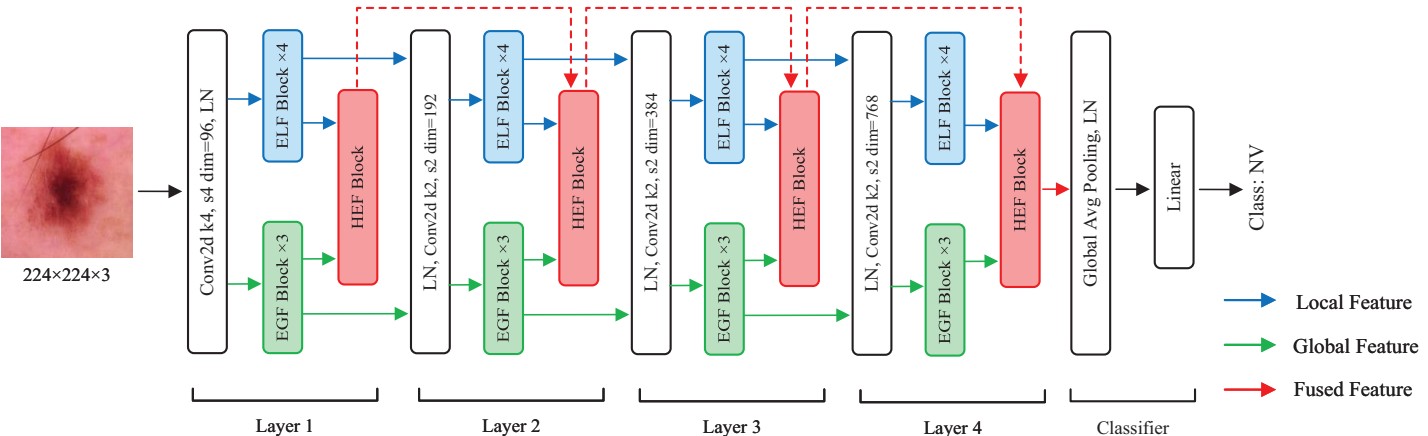

**Figure 1 The overview structure of HEMF.**

besides the local, global, and feature fusion branches, resulting in a complex model and disruption on local and global features.

To mitigate the limitations of model complexity and insufficient feature characterization capabilities, we propose an adaptive hierarchical enhanced multi-attention feature fusion framework named HEMF. We use enhanced local feature (ELF) and enhanced global feature (EGF) extraction modules to hierarchically and parallelly extract multi-scale features. We also design a hierarchical enhanced feature (HEF) fusion module, which includes Mixed Attention mechanism and Squeezed Inverted Residual Multi-Layer Perceptron (SIRMLP). The proposed HEF is capable of fusing local and global representations from various scales, learning the relation of different samples and high-dimensional representations of the fused features.

## METHODOLOGY

### Overview

In this section, we describe the details of HEMF. As depicted in Fig. 1, HEMF is built upon three distinct branches: a local branch (marked with blue), dedicated to extracting enhanced local features (ELF); a global branch (marked with green), designed to extract enhanced global features (EGF); and a feature fusion branch (marked with red), to fuse the hierarchical features obtained from local and global branches.

The model takes as input an image of size 224 × 224 × 3. Initially, the image is convolved using a convolutional kernel with a stride of 4 and a kernel size of 4, and increasing to 96 channels. The architecture subsequently employs parallel four-layer local and global feature extraction branches, each dedicated to capturing the distinct scale-specific representations. Within each layer of the local and global branches, the feature maps first undergo a layer normalization (LN) before being fed into four stacked ELF extraction blocks and three stacked EGF extraction blocks, respectively. We use four ELF extraction blocks and three EGF extraction blocks in each layer to achieve a characteristic balance of different scales. In the feature fusion branch, the hierarchical enhanced feature (HEF) fusion module of each layer receives the outputs from the

**Figure 2** The structure of ELF and EGF extraction module in HEMF compare with designs in HiFuse.

corresponding ELF and EGF extraction blocks, as well as the output from the previous HEF layer, to perform enhanced feature fusion. The fusion output serves as the input for the next HEF block. After each layer, the resolution of the feature map is reduced to 1/4, while the number of channels is doubled. Following the four-layer feature fusion, the resulting $7 \times 7 \times 768$ feature map sequentially undergoes a global average pooling, a layer normalization, and finally a linear layer to obtain the classification result.

## Enhanced local feature extraction module

The enhanced local feature (ELF) extraction module is designed to extract local detailed features from the image, which enables the model to understand the fine-grained details of the samples and plays a crucial role in image classification. As illustrated on the right of Fig. 2, the ELF extraction module comprises a $3 \times 3$ standard convolution, a $3 \times 3$ depthwise convolution, and a $1 \times 1$ pointwise convolution. The input feature map undergoes a $3 \times 3$ standard convolution, followed by a layer normalization. It sequentially undergoes the $3 \times 3$ depthwise convolution and the $1 \times 1$ pointwise convolution. After applying the Gaussian Error Linear Unit (GELU) activation function, the resulting feature map is added to the original input feature map *via* a skip connection. Compared with the local feature block in HiFuse (*Huo et al., 2024*), a $3 \times 3$ standard convolution is introduced, as seen in the left of Fig. 2. The ELF extraction module employs the depthwise and pointwise convolutions to reduce computational cost while utilizing standard convolution to enhance local feature characterization capability. The skip connection retains detailed

spatial and semantic information from the input feature map. The ELF extraction module is formulated as follows:

$$l_i = \mathrm{LN}(f^{3\times3}(L_{i-1})) \tag{1}$$

$$L_i = \mathrm{GELU}(f^{1\times1}(f^{\mathrm{depth}3\times3}(l_i))) + L_{i-1} \tag{2}$$

where $f^{n\times n}$ is the convolution operation with a kernel size of $n \times n$, $\mathrm{LN}(\cdot)$ indicates layer normalization operation, $L_{i-1}$ and $L_i$ indicate the input and output of the ELF block, respectively.

## Enhanced global feature extraction module

The enhanced global feature (EGF) extraction module is designed to extract global semantic features from the image, which facilitates the model to comprehend the global semantic information of the samples. Due to the significant intra-class variability and inter-class similarity often observed in medical images, global features are crucial for medical image classification.

As a powerful computer vision architecture, ViT captures global semantic information by employing sophisticated spatial transformations and learning long-range feature dependencies. However, as the core module of ViT, the self-attention mechanism lacks explicit spatial prior information and exhibits high quadratic complexity. To improve the performance of ViT, Retentive Networks Meet Vision Transformers (RMT) (*Fan et al., 2024*) proposes a novel self-attention mechanism called Manhattan Self-Attention (MaSA). MaSA transforms the unidirectional and one-dimensional temporal decay into bidirectional and two-dimensional spatial decay, introducing rich and explicit spatial prior information into ViT and reducing the computational complexity to linear. The Manhattan Self-Attention can be formulated as follows:

$$\mathrm{MaSA}(X) = (\mathrm{Softmax}(QK^{\mathrm{T}}) \odot D^{2d})V \tag{3}$$

$$D^{2d}_{nm} = \gamma^{|x_n - x_m| + |y_n - y_m|} \tag{4}$$

where $Q$, $K$ and $V$ indicate the query matrix, key matrix and value matrix, respectively. $\odot$ indicates the hadamard product, $\gamma$ indicates the decay value, and $(x_n, y_n)$ indicates the unique two-dimensional position coordinate within the plane of the $n$-th token.

Inspired by RMT, we propose a transformer-like enhanced global feature (EGF) extraction module. As illustrated on the right of Fig. 2, the EGF extraction module consists of a position embedding block, an RMT block, as well as a linear layer. The input feature map undergoes once position embedding (only once even with multiple stacked EGF blocks) and is then fed into the RMT block. After a skip connection and layer normalization, the output is passed to the linear layer, followed by another skip connection and layer normalization. Compared with the global feature block in HiFuse (*Huo et al., 2024*), we add skip connections with layer normalization and use RMT block instead of the Window-based Multi-head Self-Attention (W-MSA) (*Liu et al., 2021*) to enhance global feature characterization capability, as seen in the left of Fig. 2. The EGF extraction module leverages the RMT block to capture global semantic information and long-range feature dependencies, while the linear layer enhances global feature characterization. The two skip

connections combined with layer normalization preserve detailed spatial and semantic information from the input data, mitigating overfitting to some extent. The EGF extraction module is formulated as follows:

$$g_i = \text{LN}(\text{RMT}(\text{PosEmb}(G_{i-1})) + \text{PosEmb}(G_{i-1})) \qquad (5)$$

$$G_i = \text{LN}(\text{Linear}(g_i) + g_i) \qquad (6)$$

where $G_{i-1}$ and $G_i$ indicate the input and output of the EGF block, respectively.

## Hierarchical enhanced feature fusion module

Feature fusion contributes to enhance model performance and robustness by integrating features from diverse sources, hierarchical levels, or modalities (*Ngiam et al., 2011*). To fuse the enhanced local and global features, as well as utilize the high-dimensional fused features at each layer, we propose a hierarchical enhanced feature (HEF) fusion module with the similar three-branch structure of HiFuse (*Huo et al., 2024*). The core components of the module are a novel attention mechanism and an inverted residual block named Mixed Attention and Squeezed Inverted Residual Multi-Layer Perceptron (SIRMLP), respectively. The HEF fusion module can adaptively fuse the enhanced local and global features from the same layer and the fused features from the preceding layer. As shown in Fig. 3, the module consists of a local branch, a global branch, and a main branch. In the local and global branches, the input feature maps undergo the Mixed Attention, which can capture the relation of different samples and focus on critical features and spatial regional information. In the main branch, the input feature map undergoes operations such as convolution, average pooling, and concatenation. The resultant feature map undergoes the SIRMLP, which can efficiently learn the high-dimensional representation of the fused features with a low cost. The HEF fusion module is formulated as follows:

$$h_x = \text{AvgPool}(f^{1\times1}_{\text{dim}\times2}(HEF_{i-1})) \qquad (7)$$

$$h_y = \text{GELU}(f^{1\times1}(\text{LN}(\text{Concat}[h_x, f^{1\times1}(L_i), f^{1\times1}(G_i)]))) \qquad (8)$$

$$HEF_i = \text{SIRMLP}(\text{LN}(\text{Concat}[\text{MixAttn}(G_i), \text{MixAttn}(L_i), h_y])) + h_x \qquad (9)$$

where $L_i$ and $G_i$ indicate the output of multiple stacked ELF and EGF blocks, respectively. $HEF_{i-1}$ indicates the output of the previous HEF layer and $HEF_i$ indicates the output of the fusion. Additionally, as a general module, the proposed HEF fusion module can be further adopted in other downstream tasks for efficient feature fusion.

### Mixed Attention mechanism

The attention mechanism was first proposed by *Bahdanau, Cho & Bengio (2014)*. It is an adaptive selection process that mimics the selective focus of human information processing. It allows models to dynamically adjust their attention weights, highlighting important information and suppressing irrelevant information. The Spatial Attention (SA) mechanism enables the network to focus on salient spatial regions, while the Channel Attention (CA) mechanism enables the network to focus on significant features. Previous studies have demonstrated the effectiveness of combining these two attention mechanisms (*Woo et al., 2018*; *Chen et al., 2017*). The External Attention (EA) mechanism enables the

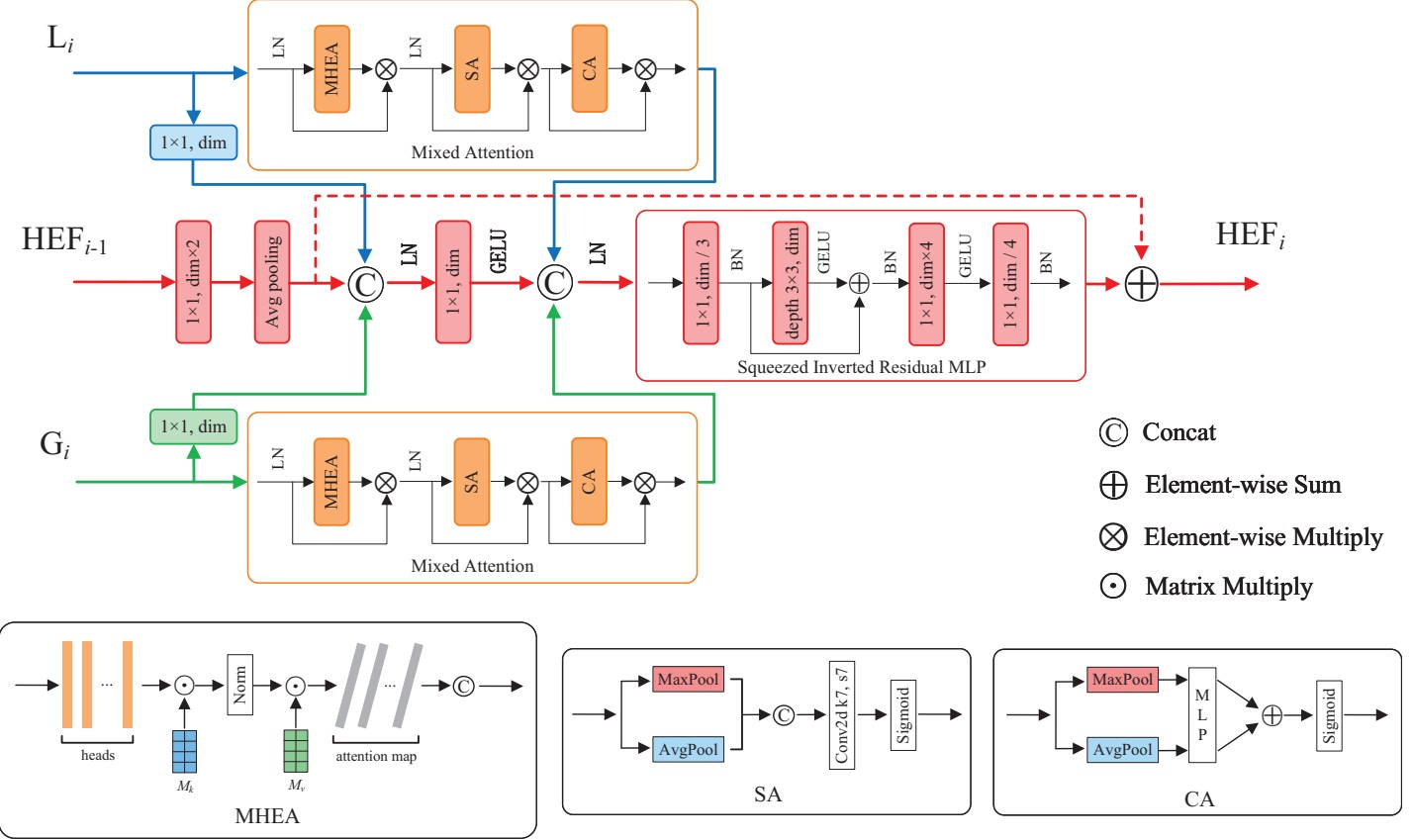

**Figure 3** The structure of HEF fusion module.

network to capture the relation of different samples, and the Multi-Head External Attention (MHEA) mechanism extends this capability, allowing the model to simultaneously capture multiple salient feature relation of different samples (*Guo et al., 2022*). SA, CA, EA, MHEA can be formulated as:

$$SA(x) = \text{Sigmoid}(f^{7 \times 7}(\text{Concat}[\text{AvgPool}(x), \text{MaxPool}(x)])) \tag{10}$$

$$CA(x) = \text{Sigmoid}(\text{MLP}(\text{AvgPool}(x)) + \text{MLP}(\text{MaxPool}(x))) \tag{11}$$

$$EA(x) = \text{Norm}(xM_k^{\text{T}})M_v \tag{12}$$

$$MHEA(x) = \text{Concat}[EA(x_1), \ldots, EA(x_H)], \quad x = \text{Concat}[x_1, x_2, \ldots, x_H] \tag{13}$$

where $M_k$ and $M_v$ are external learnable parameters. The Norm($\cdot$) indicates Double-Normalization proposed by *Guo et al. (2021)*, and $H$ indicates the number of attention heads.

To enable our model to simultaneously learn multiple salient feature relation of different samples, as well as further focus on important features and salient spatial regions, we propose a novel attention mechanism named Mixed Attention. As illustrated in Fig. 3, the Mixed Attention consists of a MHEA, an SA and a CA, along with layer normalization and multiplicative skip connections. The Mixed Attention is formulated as follows:

$$m_x = \text{LN}(\text{MHEA}(\text{LN}(M_{in})) \times \text{LN}(M_{in})) \tag{14}$$

$$m_y = m_x \times \text{SA}(m_x) \tag{15}$$

$$M_{out} = m_y \times \text{CA}(m_y) \tag{16}$$

where $M_{in}$ and $M_{out}$ indicate the input and output of the Mixed Attention module, respectively.

### Squeezed Inverted Residual Multi-Layer Perceptron

The Inverted Residuals was first proposed in MobileNetV2 (*Sandler et al., 2018*). It first employs lightweight expansion convolutions to increase the channels of the input feature map, followed by a depthwise separable convolutions for feature extraction and a $1 \times 1$ convolution to reduce the channels. Skip connections between the input and output of the inverted residual block maintain the information flow. An inverted bottleneck structure is established for a smaller number of input and output channels and a larger intermediate convolutional layer. This allows for effective learning of high-dimensional representations of the intermediate features.

The feature fusion module called HFF in HiFuse (*Huo et al., 2024*) employs an inverted residual multi-layer perceptron (IRMLP) to learn high-dimensional representations of intermediate features. It consists of a $3 \times 3$ depthwise convolution with skip connection, a pointwise convolution for expanding channels, a linear layer, and a GELU activation function. Through experimentation, we observe that when IRMLP is applied to multi-scale feature fusion, its high-dimensional input originating from three branches results in a substantial module parameters and FLOPs when expanding channels, as detailed in Table 3.

To reduce computational complexity and enable effective intermediate feature fusion and high-dimensional representation learning, we present SIRMLP, a novel inverted residual architecture. As depicted in Fig. 3, SIRMLP initially applies a pointwise convolution to reduce the input channels. Subsequently, an inverted residual structure similar to IRMLP is implemented, consisting of a $3 \times 3$ depthwise convolution, a pointwise convolution, and a linear layer. This design maintains low parameters and FLOPs during channel expansion. Table 3 presents a comparison of the module parameters and FLOPs between SIRMLP and IRMLP in four-layer feature fusion, demonstrating that the SIRMLP achieves a reduction of approximately 84.5% in both parameters and FLOPs. The SIRMLP is formulated as:

$$s_x = \text{BN}(f_{\dim/3}^{1\times1}(S_{in})) \tag{17}$$

$$s_y = \text{BN}(\text{GELU}(f^{\text{depth}3\times3}(s_x)) + s_x) \tag{18}$$

$$S_{out} = \text{BN}(f_{\dim/4}^{1\times1}(\text{GELU}(f_{\dim\times4}^{1\times1}(s_y)))) \tag{19}$$

where $\text{BN}(\cdot)$ indicates batch normalization operation, $S_{in}$ and $S_{out}$ indicate the input and output of SIRMLP, respectively.

**Table 3 Module parameters and FLOPs of SIRMLP (SIR) and IRMLP (IR) in each layer.**

| Layer | Input size: [channels, height, width] | Parameters (M) | | | FLOPs (G) | | |
|---|---|---|---|---|---|---|---|
| | | SIR | IR | Decline (%) | SIR | IR | Decline (%) |
| 1 | [96 × 3, 56, 56] | 0.185 | 1.190 | 84.450 | 0.582 | 3.733 | 84.419 |
| 2 | [192 × 3, 28, 28] | 0.739 | 4.757 | 84.473 | 0.580 | 3.731 | 84.458 |
| 3 | [384 × 3, 14, 14] | 2.952 | 19.025 | 84.484 | 0.579 | 3.729 | 84.477 |
| 4 | [768 × 3, 7, 7] | 11.802 | 76.093 | 84.490 | 0.578 | 3.729 | 84.487 |

# EXPERIMENTS

In this section, we evaluate the performance of the HEMF model on three real-world datasets including ISIC2018, Kvasir and COVID-19 CT datasets. Diverse datasets facilitate a more comprehensive assessment of model generalization performance. During the training phase, the model takes medical images and the corresponding labels as input. In the inference phase, the model receives medical images as input and outputs predicted disease categories. We employ a comprehensive set of evaluation metrics to assess model performance and conduct ablation studies to verify the effectiveness of individual model components. Finally, we provide heatmap visualization to further demonstrate the effectiveness of HEMF.

## Datasets

### ISIC2018 dataset

The ISIC2018 dataset (*Codella et al., 2019*) consists of 10, 015 images from seven categories of skin lesions, including melanocytic nevi (NV), dermatofibroma (DF), melanoma (MEL), actinic keratosis (AKIEC), benign keratosis (BKL), basal cell carcinoma (BCC) and vascular lesions (VASC). Due to its inter-class similarity and data imbalance, this dataset serves as an important benchmark for medical image classification tasks. The specific class distribution is detailed in Table 4. Representative image samples from the ISIC2018 dataset are displayed in Fig. 4A. The original size of these images is 650 × 450 pixels. For our experiments, we resize all the images to 224 × 224 pixels. Following the data partition strategy of HiFuse (*Huo et al., 2024*) and ResGANet (*Cheng et al., 2022*), we allocate 70% of the samples for training and validation, while the remaining 30% for testing.

### Kvasir dataset

The Kvasir dataset (*Pogorelov et al., 2017*) comprises 4,000 images across eight endoscopic gastrointestinal diseases, annotated and validated by experienced endoscopists. These categories encompass three anatomical landmarks (Z-line, pylorus, and cecum), three pathological findings (esophagitis, polyps, and ulcerative colitis), and two categories related to polyp removal (dyed and lifted polyps and dyed resection margins). Each category contains 500 images, as detailed in Table 4. Figure 4B shows the example images of Kvasir. We choose this dataset because it is of great significance to computer aided gastrointestinal disease detection and widely adopted in medical analysis research. The size of the original images varies ranging from 720 × 576 to 1,920 × 1,072 pixels. To facilitate

**Table 4 Data distribution of ISIC2018, Kvasir and COVID-19 CT datasets.**

| ISIC2018 | | Kvasir | | COVID-19 CT | |
|---|---|---|---|---|---|
| NV | 6,075 | Z-line | 500 | COVID | 349 |
| MEL | 1,113 | Pylorus | 500 | NonCOVID | 397 |
| BKL | 1,099 | Cecum | 500 | | |
| BCC | 514 | Esophagitis | 500 | | |
| AKIEC | 327 | Polyps | 500 | | |
| VASC | 142 | Ulcerative colitis | 500 | | |
| DF | 115 | Dyed and lifted polyps | 500 | | |
| | | Dyed resection margins | 500 | | |
| Seven classes | 10,015 | Eight classes | 4,000 | Two classes | 746 |

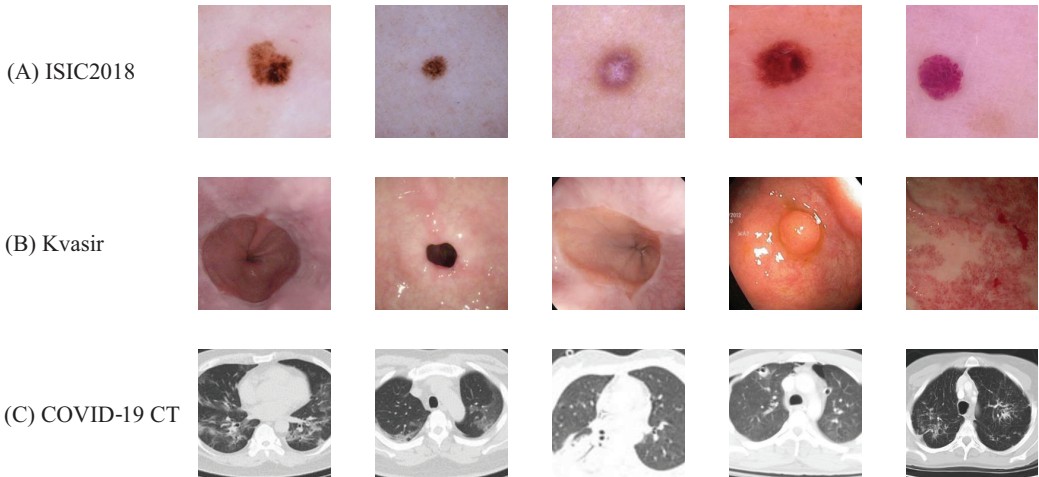

**Figure 4 Sample images from the (A) ISIC2018, (B) Kvasir and (C) COVID-19 CT datasets with standardized preprocessing.** All images are resized to 224 × 224 pixels for model input consistency.

subsequent experiments, we resize all images to 224 × 224 pixels. Following the data partition strategy *Huo et al. (2024)* and *Pogorelov et al. (2017)*, we divide the dataset into training and test sets with a 0.5 : 0.5 ratio with two-fold cross-validation.

### COVID-19 CT dataset

The COVID-19 CT dataset (*He et al., 2020*) comprises 746 CT scan images, with 349 images positive for COVID-19 and 397 images negative or indicative of other diseases (NonCOVID), as detailed in Table 4. Figure 4C shows several image samples of the dataset. We choose this dataset because it is widely adopted and CT scans are promising in providing accurate, fast, and cheap screening and testing different types of diseases. The size of the images varies from 143 × 76 to 1,637 × 1,225 pixels. For our experiments, we resized all the images to 224 × 224 pixels. Following the data partition strategy *Huo et al. (2024)* and *He et al. (2020)*, we divide the dataset into training, validation, and test sets with a ratio of 0.6:0.15:0.25, respectively.

## Metrics

We use accuracy (Acc), precision (Prec), recall, F1-score, Matthews Correlation Coefficient (MCC), kappa, Receiver Operating Characteristic (ROC) curve, and the area under the ROC curve (AUC) as evaluation metrics. Here, accuracy represents the proportion of correctly classified instances out of the total number of instances. Precision measures the correct predicted proportion of all instances predicted as positive. Recall quantifies the proportion of actual positive instances that are correctly identified. F1-score is the harmonic mean of precision and recall, with higher values indicating better classification performance. The Matthews Correlation Coefficient (MCC) is a metric for evaluating classification performance in binary classification tasks. Its value ranges from −1 to +1, with higher values indicating better classification performance. MCC incorporates all four confusion matrix categories to provide a balanced performance measure, making it particularly suitable for imbalanced datasets. Kappa coefficient is used to measure the consistency of classifiers. The value of Kappa coefficient ranges from 0 to 1, where 1 indicates perfect consistency and 0 indicates random consistency. The ROC curve is a plot with the false positive rate (FPR) on the x-axis and the true positive rate (TPR) on the y-axis, which is less affected by imbalance class distributions. A larger area under the ROC curve (AUC) indicates superior classification performance. These metrics are formulated as follows:

$$Acc = \frac{TP + TN + FP + FN}{TP + TN} \tag{20}$$

$$Prec = \frac{TP}{TP + FP} \tag{21}$$

$$Recall = TPR = \frac{TP}{TP + FN} \tag{22}$$

$$F1 = \frac{2 \times (Prec + Recall)}{Prec \times Recall} \tag{23}$$

$$MCC = \frac{TP \times TN - FP \times FN}{\sqrt{(TP + FP)(TP + FN)(TN + FP)(TN + FN)}} \tag{24}$$

$$kappa = \frac{P_o - P_e}{1 - P_e} \tag{25}$$

$$FPR = \frac{FP}{FP + TN} \tag{26}$$

$$AUC = \sum_{i=1}^{n-1} \frac{TPR_i + TPR_{i+1}}{2} \times (FPR_i + FPR_{i+1}) \tag{27}$$

where *TP*, *TN*, *FP* and *FN* indicate true positive, true negative, false positive, and false negative, respectively. $P_o$ represents the observed agreement rate (the empirical classification accuracy), and $P_e$ denotes the expected chance agreement rate (the accuracy achievable through random predictions).

**Table 5 Experimental setting.**

| Training config | |
|---|---|
| Initial learning rate | 1e−4 |
| Final learning rate | 1e−6 |
| Optimizer | AdamW |
| Learning rate schedule | CosineAnnealing |
| Warm up schedule | Linear |
| Warm up epochs | 1 |
| Weight decay | 0.01 |
| Epochs | 150 |
| Batch size | 32 |
| Loss function | Categorical cross-entropy |
| Drop path rate | 0 |

## Implementation details

The experiments are conducted on the Ubuntu 20.04.6 LTS operating system. GPU acceleration is provided by an NVIDIA Tesla T4 with 16 GB memory. The software environment includes Python 3.9, PyTorch 1.13.1, and CUDA 11.7.

The experimental parameters are set as follows. The initial and final learning rates are 1e−4 and 1e−6, respectively. AdamW optimizer with the cosine annealing learning rate scheduling strategy is adopted. We train 150 epochs with a batch size of 32. We use the categorical cross-entropy (CCE) loss function defined by the following formula:

$$L = -\frac{1}{N}\sum_{n=1}^{N}\sum_{i=1}^{C} y_i \log \hat{y}_i \tag{28}$$

where $N$ represents the number of samples, $C$ represents the number of classes, $y_i$ is the true label for sample $i$, and $\hat{y}_i$ is the predicted label for sample $i$. To accelerate training, mixed-precision training is utilized. Our code can be available from https://github.com/Esgjgd/HEMF. More parameter settings are detailed in Table 5.

## Results

To evaluate the classification performance, we compare HEMF with advanced models in recent years, including Conformer (*Peng et al., 2021*), ConvNext (*Liu et al., 2022*), PerViT (*Min et al., 2022*), Focal (*Yang et al., 2022*), UniFormer (*Li et al., 2023*), BiFormer (*Zhu et al., 2023*), HiFuse (*Huo et al., 2024*) and CAME (*Li, 2023*). We conduct experiments on the ISIC2018, Kvasir and COVID-19 CT datasets using identical experimental settings.

### Results on ISIC2018 dataset

Following the data partitioning strategy in HiFuse (*Huo et al., 2024*) and ResGANet (*Cheng et al., 2022*), we train our models from scratch, allocating 70% of the samples for training and validation and 30% for testing. As shown in Table 6, HEMF achieves the highest accuracy (87.34%) with only 66.64M parameters and 10.01G FLOPs. Notably, HEMF demonstrates significant improvements over the previous best model HiFuse_Base.

**Table 6 Performance comparison on ISIC2018 dataset.** Bold entries indicate the optimal results, whereas underlined entries denote the second-best results.

| Method | Params (M) | FLOPs (G) | Acc (%) | F1 (%) | Prec (%) | Recall (%) |
|---|---|---|---|---|---|---|
| Conformer-base-p16 (*Peng et al., 2021*) | 83.29 | 22.89 | 82.66 | 72.44 | 73.31 | 71.66 |
| ConvNeXt-B (*Liu et al., 2022*) | 88.59 | 15.36 | 79.95 | 63.24 | 64.90 | 62.06 |
| PerViT-M (*Min et al., 2022*) | 43.04 | 9.00 | 81.64 | 67.66 | 68.19 | 67.29 |
| Focal-B (*Yang et al., 2022*) | 87.10 | 15.30 | 79.64 | 62.88 | 65.73 | 60.68 |
| UniFormer-B (*Li et al., 2023*) | 50.02 | 8.30 | 82.44 | 68.41 | 70.67 | 66.54 |
| BiFormer-B (*Zhu et al., 2023*) | 56.04 | 9.80 | 82.66 | 68.95 | 72.66 | 66.47 |
| HiFuse_Tiny (*Huo et al., 2024*) | 82.49 | 8.13 | 82.99 | 72.99 | 73.67 | 72.87 |
| HiFuse_Small (*Huo et al., 2024*) | 93.82 | 8.84 | 83.59 | 72.70 | 72.70 | 73.14 |
| HiFuse_Base (*Huo et al., 2024*) | 127.80 | 10.97 | 85.85 | 75.32 | 74.57 | 76.58 |
| CAME (*Li, 2023*) | 106.38 | 11.63 | 84.17 | 74.43 | 74.08 | 74.98 |
| HEMF (ours) | 66.64 | 10.01 | **87.34** | **78.89** | **80.42** | **77.59** |

**Table 7 Performance comparison on Kvasir dataset.** Bold entries indicate the optimal results, whereas underlined entries denote the second-best results.

| Method | Acc (%) | F1 (%) | Prec (%) | Recall (%) |
|---|---|---|---|---|
| Conformer-base-p16 (*Peng et al., 2021*) | 84.25 | 84.27 | 84.45 | 84.37 |
| ConvNeXt-B (*Liu et al., 2022*) | 74.62 | 74.41 | 75.69 | 74.62 |
| PerViT-M (*Min et al., 2022*) | 82.40 | 82.30 | 82.88 | 82.40 |
| Focal-B (*Yang et al., 2022*) | 78.00 | 77.93 | 78.19 | 78.01 |
| UniFormer-B (*Li et al., 2023*) | 83.10 | 83.04 | 83.09 | 83.10 |
| BiFormer-B (*Zhu et al., 2023*) | 84.25 | 84.26 | 84.67 | 84.25 |
| HiFuse_Tiny (*Huo et al., 2024*) | 84.85 | 84.89 | 84.96 | 84.90 |
| HiFuse_Small (*Huo et al., 2024*) | 86.12 | 86.13 | 86.25 | 86.13 |
| HiFuse_Base (*Huo et al., 2024*) | 85.97 | 86.07 | 86.29 | 86.01 |
| CAME (*Li, 2023*) | 85.68 | 85.68 | 85.77 | 85.68 |
| HEMF (ours) | **87.03** | **87.02** | **87.34** | **87.03** |

Specifically, HEMF reduces the number of model parameters by 47.86% and FLOPs by 8.75% while improving 1.74% in accuracy, 7.84% in precision, 1.32% in recall, and 4.74% in F1-score.

### Results on Kvasir dataset

Following the data partitioning strategy in HiFuse (*Huo et al., 2024*) and (*Pogorelov et al., 2017*), we train our models from scratch, dividing the dataset into training and test sets with a 0.5:0.5 ratio. We performed a two-fold cross-validation experiment and reported the average results. As shown in Table 7, HEMF achieves the highest accuracy (87.03%). Compared to the previous best model HiFuse_Small, HEMF achieves improvements of 1.06% in accuracy, 1.26% in precision, 1.04% in recall, and 1.03% in F1-score.

**Table 8 Performance comparison on COVID-19 CT dataset.** Bold entries indicate the optimal results, whereas underlined entries denote the second-best results.

| Method | Acc (%) | F1 (%) | Prec (%) | Recall (%) |
|---|---|---|---|---|
| Conformer-base-p16 (*Peng et al., 2021*) | 75.81 | 75.60 | 76.81 | 77.81 |
| ConvNeXt-B (*Liu et al., 2022*) | 55.38 | 54.68 | 54.95 | 54.81 |
| PerViT-M (*Min et al., 2022*) | 74.75 | 73.95 | 75.96 | 73.93 |
| Focal-B (*Yang et al., 2022*) | 69.36 | 68.06 | 70.66 | 68.35 |
| UniFormer-B (*Li et al., 2023*) | 71.38 | 71.35 | 72.18 | 71.89 |
| BiFormer-B (*Zhu et al., 2023*) | 72.05 | 71.95 | 71.94 | 71.96 |
| HiFuse_Tiny (*Huo et al., 2024*) | 74.73 | 74.67 | 74.65 | 74.73 |
| HiFuse_Small (*Huo et al., 2024*) | 76.88 | 76.31 | 77.78 | 76.19 |
| HiFuse_Base (*Huo et al., 2024*) | 76.34 | 76.17 | 76.30 | 76.11 |
| CAME (*Li, 2023*) | 78.49 | 78.47 | 78.49 | 78.61 |
| HEMF (ours) | **82.26** | **82.20** | **82.18** | **82.22** |

## Results on COVID-19 CT dataset

Following the data partitioning strategy in HiFuse *Huo et al. (2024)* and *He et al. (2020)*, we train our models from scratch, allocating 60% of the data for training, 15% for validation, and 25% for testing. As shown in Table 8, HEMF achieves the highest accuracy (82.26%). Compared to the previous best model CAME, HEMF achieves improvements of 4.80% in accuracy, 4.70% in precision, 4.59% in recall, and 4.75% in F1-score, demonstrating a significant improvement.

## ROC curves and confusion matrices

Figure 5 illustrates the ROC curves and confusion matrices for HEMF on the ISIC2018, Kvasir and COVID-19 CT datasets. As shown, HEMF exhibits remarkable classification performance across all three datasets. On the ISIC2018 dataset, despite the similarity between MEL and NV, HEMF can effectively distinguish them. On the Kvasir dataset, HEMF can validly differentiates two similar types of dyed and lifted polyps and dyed resection margins.

To further validate consistency, we evaluate the MCC and kappa coefficient on ISIC2018 and Kvasir datasets. Table 9 shows that HEMF outperforms other models in both metrics, demonstrating robust classification consistency.

## Ablation study

To evaluate the impact of each key component of HEMF and gain a deeper comprehension, we conduct an ablation study on the ISIC2018 dataset, examining the contributions of the enhanced local feature (ELF) block, enhanced global feature (EGF) block, Multi-Head External Attention (MHEA), Channel and Spatial Attention (CA&SA), and Squeezed Inverted Residual Multi-Layer Perceptron (SIRMLP). As shown in Table 10, metrics are relatively low when only the ELF block or EGF block is used. The recall improves significantly when both ELF and EGF blocks are employed. Adding MHEA or CA&SA results in a moderate increase in accuracy. Further improvements in F1-score,

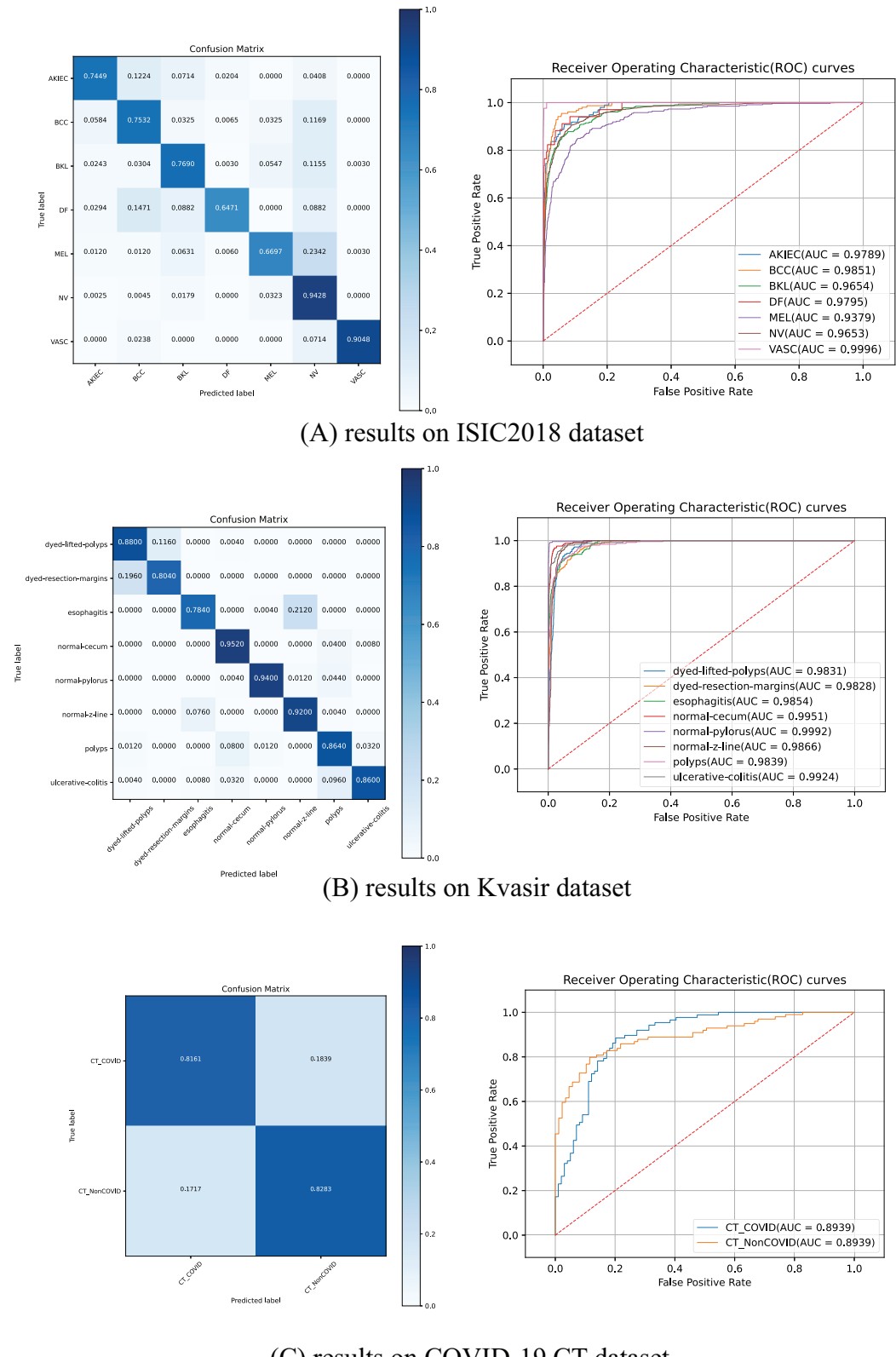

(A) results on ISIC2018 dataset

(B) results on Kvasir dataset

(C) results on COVID-19 CT dataset

**Figure 5  The ROC curves and confusion matrices of HEMF on different datasets.**

**Table 9 MCC and kappa comparison on ISIC2018 and Kvasir datasets.** Bold entries indicate the optimal results, whereas underlined entries denote the second-best results.

|  | ISIC2018 | | Kvasir | |
|---|---|---|---|---|
|  | MCC | Kappa | MCC | Kappa |
| ConvNeXt-B (*Liu et al., 2022*) | 0.6022 | 0.6005 | 0.7119 | 0.7100 |
| PerViT-M (*Min et al., 2022*) | 0.6412 | 0.6406 | 0.7998 | 0.7989 |
| Focal-B (*Yang et al., 2022*) | 0.5964 | 0.5949 | 0.7490 | 0.7486 |
| UniFormer-B (*Li et al., 2023*) | 0.6505 | 0.6480 | 0.8070 | 0.8069 |
| BiFormer-B (*Zhu et al., 2023*) | 0.6604 | 0.6592 | 0.8205 | 0.8200 |
| HiFuse_Tiny (*Huo et al., 2024*) | 0.6639 | 0.6619 | 0.8055 | 0.8051 |
| HiFuse_Small (*Huo et al., 2024*) | 0.6896 | 0.6892 | 0.8416 | 0.8414 |
| HiFuse_Base (*Huo et al., 2024*) | 0.7282 | 0.7280 | 0.8402 | 0.8400 |
| CAME (*Li, 2023*) | 0.6967 | 0.6960 | 0.8364 | 0.8363 |
| HEMF (ours) | **0.7553** | **0.7551** | **0.8459** | **0.8580** |

**Table 10 Component ablation study on ISIC2018 dataset.**

| Key components | | | | | Acc (%) | F1 (%) | Prec (%) | Recall (%) |
|---|---|---|---|---|---|---|---|---|
| ELF | EGF | MHEA | CA&SA | SIRMLP | | | | |
| ✓ |  |  |  |  | 81.17 | 64.70 | 66.23 | 63.90 |
|  | ✓ |  |  |  | 81.84 | 66.61 | 65.56 | 69.20 |
| ✓ | ✓ |  |  |  | 82.14 | 70.06 | 73.33 | 68.27 |
| ✓ | ✓ | ✓ |  |  | 84.04 | 70.80 | 72.98 | 69.41 |
| ✓ | ✓ |  | ✓ |  | 83.41 | 70.30 | 70.98 | 69.92 |
| ✓ | ✓ | ✓ | ✓ |  | 84.54 | 72.87 | 72.97 | 72.81 |
| ✓ | ✓ | ✓ | ✓ | ✓ | 87.34 | 78.89 | 80.42 | 77.59 |

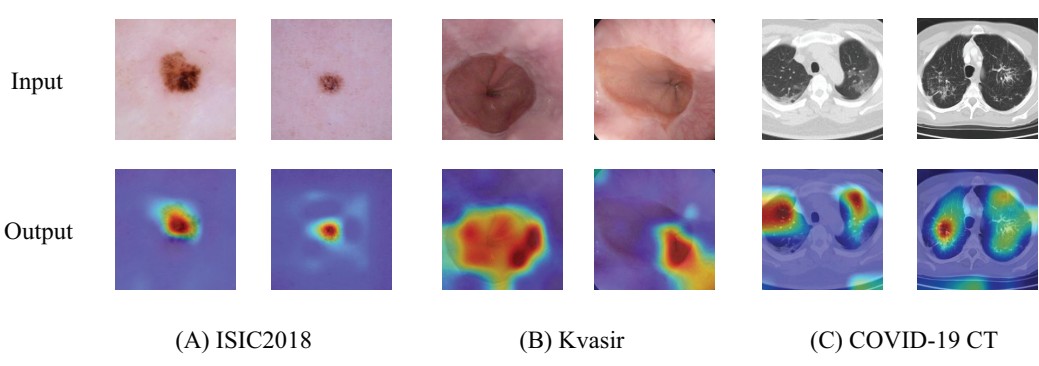

Input

Output

(A) ISIC2018      (B) Kvasir      (C) COVID-19 CT

**Figure 6 Heatmap visualization of (A) ISIC2018, (B) Kvasir and (C) COVID-19 CT datasets.**

precision, and recall are observed when all three attention mechanisms are used together (Mixed Attention). Finally, the addition of SIRMLP (*i.e.*, the complete HEMF model) leads to a substantial improvement. We believe that SIRMLP can effectively learn and leverage

the high-dimensional representation of the fused features, significantly enhancing the characterization capability of HEMF.

### Heatmap visualization

To further validate the capability of HEMF in capturing discriminative features, we employed Grad-CAM++ (*Chattopadhay et al., 2018*) to generate heatmaps by visualizing the output of the final layer (excluding the linear layer). Figure 6 presents representative visualization results across three medical image datasets: (A) ISIC2018, (B) Kvasir, and (C) COVID-19 CT. The generated heatmaps demonstrate that HEMF consistently localizes pathological regions with high precision, indicating its effectiveness in hierarchically integrating both global contextual information and local discriminative features. Multi-scale fusion mechanism enables the model to reliably identify and characterize lesion patterns across diverse medical image modalities.

### Limitations

The proposed HEMF model achieves state-of-the-art performance in medical image classification, underscoring the promise of hierarchical multi-scale feature extraction and fusion. However, several potential limitations should be noted:

- While HEMF demonstrates superior model parameter efficiency and computational economy relative to existing approaches, further architectural refinements could potentially enhance its compactness.
- The current HEMF model employs the conventional categorical cross-entropy loss function. Exploring alternative or adaptive loss functions may lead to further performance improvements, particularly in response to varying dataset distributions.
- The current HEMF implementation does not incorporate any data augmentation techniques such as CutMix and MixUp. Integrating advanced augmentation strategies could enhance model generalization, improve robustness to perturbations, and mitigate class imbalance.

## CONCLUSION

In this article, we propose HEMF, an adaptive hierarchical enhanced multi-attention feature fusion framework for cross-scale medical image classification. HEMF incorporates enhanced local feature (ELF) and enhanced global feature (EGF) extraction modules, enabling the parallel extraction of multi-scale local and global features. To fully fuse and utilize local and global features from different scales, a hierarchical enhanced feature (HEF) fusion module is proposed, which includes a novel attention mechanism named Mixed Attention and a novel inverted residual block named Squeezed Inverted Residual Multi-Layer Perceptron (SIRMLP). Extensive experiments have demonstrated that HEMF achieves state-of-the-art performance on the ISIC2018, Kvasir and COVID-19 CT datasets with a relatively low model parameter. The success of HEMF further validates the significance of multi-scale local and global features and their effective fusion for medical image classification tasks.

In future work, we plan to further refine the HEMF model by exploring practical loss functions and implementing dynamic hierarchical feature selection mechanisms that can be adapted to varying input data. Furthermore, we also consider extending the application of HEMF to other downstream tasks, such as medical image segmentation and multi-modal medical image fusion, to further improve performance.

### Funding
The authors received no funding for this work.

### Competing Interests
The authors declare that they have no competing interests.

### Author Contributions
- Jingdong He conceived and designed the experiments, performed the experiments, analyzed the data, performed the computation work, prepared figures and/or tables, authored or reviewed drafts of the article, and approved the final draft.
- Qiang Shi conceived and designed the experiments, analyzed the data, prepared figures and/or tables, authored or reviewed drafts of the article, and approved the final draft.
- Jun Ma conceived and designed the experiments, analyzed the data, prepared figures and/or tables, authored or reviewed drafts of the article, and approved the final draft.
- Dacheng Shi performed the experiments, performed the computation work, authored or reviewed drafts of the article, and approved the final draft.
- Tie Min performed the experiments, performed the computation work, authored or reviewed drafts of the article, and approved the final draft.

### Data Availability
All the data and code are available at GitHub and Zenodo:

- https://github.com/Esgjgd/HEMF.

- Jingdong He. (2025). Esgjgd/HEMF: HEMF New Released (new). Zenodo. https://doi.org/10.5281/zenodo.16275426.

The original datasets are available at:

- ISIC2018: https://challenge.isic-archive.com/data/#2018. (https://doi.org/10.1038/sdata.2018.161).

- Kvasir: https://datasets.simula.no/kvasir/. (doi.org/10.1145/3083187.3083212)

- COVID-19 CT: https://www.kaggle.com/datasets/luisblanche/covidct. (https://doi.org/10.34740/kaggle/ds/584020).

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
