# Peer review of "HEMF: an adaptive hierarchical enhanced multi-attention feature fusion framework for cross-scale medical image classification"

_PeerJ Computer Science, doi:10.7717/peerj-cs.3181_

## Round 0.1 · original submission · Major Revisions

I recommend that the paper undergo a significant revision. There are numerous inconsistencies. Mandatory requests!

**PeerJ Staff Note**: Please ensure that all review, editorial, and staff comments are addressed in a response letter and that any edits or clarifications mentioned in the letter are also inserted into the revised manuscript where appropriate.

**PeerJ Staff Note**: It is PeerJ policy that additional references suggested during the peer-review process should only be included if the authors agree that they are relevant and useful.

Reviewer 1 ·

Basic reporting

The manuscript requires revisions, as it must be improved in terms of clarity, figure and form, and literature completeness to meet academic standards.

Experimental design

The manuscript requires major revisions to incorporate stronger statistical validation and to address potential overfitting and generalizability concerns through rigorous cross-validation and broader dataset evaluation.

Validity of the findings

The results are promising but require revisions to include more rigorous evaluation metrics and a thorough discussion of the study’s limitations.

Additional comments

1. Explain all abbreviations upon first mention.

2. Clearly state in the introduction what the proposed approach achieves that others do not.

3. Compare and contrast HEMF with other fusion models such as CoAtNet, TransUNet, and MedFormer.

4. Retain the thematic organization of the related work section, but expand it with deeper analysis.

5. Provide a complete block diagram or flowchart of the algorithm.

6. Clearly distinguish between training-time and inference-time behavior.

7. Justify the combination of SA, CA, and MHEA with theoretical rationale or prior work.

8. Explain the data flow within the fusion module.

9. Strengthen the theoretical foundation of the attention and fusion strategies.

10. Add bar charts and ROC-AUC curves with confidence intervals.

11. Include comparisons of training time, convergence time, and inference time.

12. Provide sample heatmaps or prediction visualizations to demonstrate attention/fusion effectiveness.

13. Read this article if applicable: https://journal.qubahan.com/index.php/qaj/article/view/686

14. Explicitly discuss the weaknesses and limitations of the proposed approach.

Reviewer 2 ·

Basic reporting

-

Experimental design

-

Validity of the findings

-

Additional comments

1. While the proposed framework demonstrates promising results across multiple datasets, the manuscript must more clearly articulate the novelty of HEMF in comparison to existing multi-attention or hierarchical fusion approaches. The claim of achieving state-of-the-art performance should be substantiated with direct benchmarking against recent peer-reviewed models.

2. The introduction is generally well-structured; however, the articulation of the research gap could be more sharply defined. It is recommended to explicitly state what specific limitations in prior transformer-based medical image classification methods HEMF addresses.

3. The related work section is basically examined from three perspectives, and the methods in the literature are examined based on convolutional neural networks, machine learning, and transformers. This section requires significant expansion. A comprehensive literature review table comparing key methods, datasets, architectures, and performance metrics should be included. This will contextualize the contribution of HEMF and highlight its relative advantages.

4. The modular breakdown of HEMF is appreciated; however, the manuscript should include architectural diagrams and flowcharts to improve clarity. Additionally, the novelty of each module should be explicitly compared to similar components in existing models.

5. The use of multiple datasets is commendable. However, the manuscript should include a justification for the selection of each dataset and discuss how dataset diversity impacts generalizability.

6. This is a critical omission: A detailed parameter table must be included, specifying all training configurations (e.g., optimizer type, learning rate, batch size, number of epochs, regularization techniques). This is essential for reproducibility.

7. The evaluation section should be expanded to include additional metrics such as Cohen’s Kappa and MCC, which are particularly important in imbalanced classification tasks. These metrics will provide a more robust assessment of model performance.

As a result, although the hierarchical enhanced multi-attention feature fusion proposed in the scope of the study has a certain originality, the sections listed above should be taken into consideration.

---

## Round 0.2 · accepted · Accept

Congratulations on your valuable contribution!

Reviewer 1 ·

Basic reporting

paper has been modified and improved.

Experimental design

No more comments.

Validity of the findings

No more comments.

Additional comments

No more comments.

Reviewer 2 ·

Basic reporting

All comments are in the last section.

Experimental design

All comments are in the last section.

Validity of the findings

All comments are in the last section.

Additional comments

Thanks for the revision. The final version of the paper, together with the responses to my comments and the corresponding revisions, is generally satisfactory.